# An Innovative Method for the Production of Yoghurt Fortified with Walnut Oil Nanocapsules and Characteristics of Functional Properties in Relation to Conventional Yoghurts

**DOI:** 10.3390/foods12203842

**Published:** 2023-10-20

**Authors:** Katarzyna Turek, Gohar Khachatryan, Karen Khachatryan, Magdalena Krystyjan

**Affiliations:** 1Department of Animal Product Processing, Faculty of Food Technology, University of Agriculture in Krakow, Mickiewicz Ave. 21, 31-120 Krakow, Poland; katarzyna.turek@urk.edu.pl; 2Department of Food Analysis and Evaluation of Food Quality, Faculty of Food Technology, University of Agriculture in Krakow, Mickiewicz Ave. 21, 31-120 Krakow, Poland; gohar.khachatryan@urk.edu.pl; 3Laboratory of Nanomaterials and Nanotechnology, Faculty of Food Technology, University of Agriculture, Balicka Street 122, 30-149 Krakow, Poland; karen.khachatryan@urk.edu.pl; 4Department of Carbohydrate Technology and Cereal Processing, Faculty of Food Technology, University of Agriculture in Krakow, Mickiewicz Ave. 21, 31-120 Krakow, Poland

**Keywords:** PUFA, yoghurt, fortification, encapsulation

## Abstract

Polyunsaturated fatty acids (PUFAs) are crucial nutrients involved in a plethora of metabolic and physiological processes. PUFAs have been extensively researched for their effects on human nutrition and health. The high demand for these fatty acids offers the possibility of adding vegetable oils to dairy products such as yoghurt. The aim of this study was to produce nano/microcapsules comprising walnut oil through exclusively natural ingredients utilised in yoghurt manufacturing. Additionally, the study tested yoghurt supplemented with PUFAs using the acquired nano/microcapsules. Chemical and physiochemical properties, microbiological analysis, rheological measurements, texture analysis, scanning electron microscope (SEM) analysis, ATR-FTIR spectroscopy, and sensory and fatty acids profile analysis were performed. A physico-chemical analysis highlighted the impact of oil addition on fat and dry matter concentration, revealing an increased quantity of said components in yoghurt after oil addition. Based on the identified parameters for potential and active acidity in the yoghurts, normal lactic fermentation was observed. Furthermore, the addition of oil was found to have an impact on the pH of the yoghurt. Microbiological analysis indicated that the incorporation of nano-encapsulated walnut oil did not have any notable effect on the abundance of determined microorganisms in the yoghurt. However, it was observed that the number of *Lactobacillus delbrueckii* ssp. *bulgaricus* increased as a result of storage. The incorporation of enclosed oil in yoghurt resulted in negligible alterations in rheological and sensory characteristics when compared with the plain variant. The addition of oil had an effect on most of the analysed fatty acids. Fortified yoghurt shows a more favourable proportion of the fatty acid groups tested (SFA, MUFA, and PUFA) and lower values of fat quality factors (AI and TI).

## 1. Introduction

Currently, there is a growing interest in health-promoting foods. The promotion of the beneficial effects of food on human health is strongly promoted by health professionals and nutritionists. Awareness of this is influencing an increase in the range of functional foods [1,2]. These foods can be dairy products containing essential fatty acids (EFAs) [3,4]. These acids are not synthesised in the human body, so they must be supplied with food. Deficiencies of EFAs in the diet can lead to the development of many diseases especially cardiovascular diseases and cancer [5,6]. Therefore, an adequate ratio of EFAs belonging to n-3 and n-6 acids is extremely important. According to FAO/WHO data, the appropriate ratio of these fatty acids supplied in the diet should be 5–10:1 [7]. Some vegetable oils show a nutritionally beneficial ratio of n-6 to n-3 fatty acids, such as walnut oil, which is also a source of polyphenols, tocopherols, and carotenoids [8].

Milk and dairy products show a low content of n-3 fatty acids, so the search has begun to find ways to meet adequate nutritional requirements due to the high consumption of these livestock products [9].

Changing the diet of dairy animals and enriching dairy products with polyunsaturated fatty acids are two effective methods leading to an increase in their content. To date, research has been carried out into improving the lipid composition of dairy products through the addition of bioactive components of vegetable oils and fats and marine animal fats. Bermúdez-Aguirre and Barbosa-Cánovas [10] used the addition of microencapsulated fish oil (OceanNutrition, Dartmouth, NS, Canada) and flaxseed oil (General Nutrition Corporation, Pittsburgh, PA, USA) to produce queso fresco, cheddar, and mozzarella cheeses, obtaining acceptable products rich in omega-3 fatty acids. Similarly, Estrada et al. [3] produced strawberry yoghurt (from LB-12 and ST-M5 culture, Chr. Hansen, Milwaukee, WI, USA) with the addition of salmon oil encapsulated in chitosan microcapsules, noting no effect of this addition on the pH and syneresis of fresh yoghurts and a slight reduction in these parameters during a one-week refrigerated storage period. In contrast, Rognlien et al. [11] studied the effect of fresh and oxidised fish oil on yoghurt quality. They found that oxidised oil was not suitable for the production of this type of fermented milk due to health risks and the ease of detecting its presence in the product even by untrained panellists. In contrast, yoghurt enriched with fresh fish oil was highly resistant to oxidation due to the antioxidant activity of the peptides released during fermentation. Eicosapentaenoic acid and docosahesaenoic acid are mainly present in marine organisms such as fish and algae, while the main sources of linoleic acid are flaxseed oil, walnut oil, borage oil, and linseed oil [12]. Foods rich in α-linolenic acid are walnuts (one gram provides about 0.09 g ALA) and flaxseeds (1 g provides about 0.3 g ALA) [13]. Goyal et al. [14] consider flaxseed oil to be the most suitable plant source of omega-3 fatty acids, but it is highly susceptible to oxidation processes due to its polyunsaturated nature. These processes lead to the formation of toxic hydroperoxides and compounds with an unpleasant taste during processing. The effective way to reduce negative changes is to produce food with the addition of flaxseed oil in emulsified, microencapsulated, or as an oil-in-water emulsion. The same authors studied the technological suitability of flaxseed oil as a food additive in the form of an emulsion with whey protein concentrate WPC-80, in order to protect the oil from oxidation processes. Comunian et al. [15] produced strawberry yoghurt with the addition of cold-pressed, microencapsulated borage oil (*E. plantagineum* L.) and a mixture of plant phytosterols. Based on the results, the authors concluded that the physicochemical, rheological, and sensory properties of the fortified yoghurt were similar to the control one (without the addition of bioactive compounds) Furthermore, the addition of the borage oil in microencapsulated form improved oxidative stability of product. Indian dahi fermented milk with mesophilic microbial culture (NCDC, Karnal, India) with the addition of microencapsulated flaxseed oil was also produced [16]. Marand et al. [17] did not use vegetable oil to increase the amount of omega-3 family acids, but successfully applied flaxseed flour in the production of yoghurt from a commercial starter culture (YC-X11, Chr. Hansen, Denmark). The fortification used has improved the physicochemical properties and the ratio of omega-6 to omega-3 acids, as well as antioxidant activity. On the other hand, there was no significant deterioration in the organoleptic properties of the enriched yoghurts. Similarly, Ozturkoglu-Budak et al. [18], used the addition of dried and ground hazelnuts, walnuts, almonds and pistachios in the production of DVS-cultured yoghurts (CH1, Chr. Hansen, Denmark). Authors found that the addition of dried walnuts improved the ratio of omega-6 to omega-3 acids to the greatest extent, while the addition of all nuts increased the abundance of characteristic microorganisms. Ilyasoglu and Yilmaz [8] used walnut pulp for the same purpose, but the researchers concluded that further research is needed on the use of flavouring and thickening agents in the production of this type of yoghurt to improve the sensory acceptability of yoghurt enriched in polyunsaturated fatty acids.

Two research papers have been written on the addition of vegetable oils to kefirs. Particularly in the work on the addition of walnut and linseed oil to kefirs, no effect of the additive used on the pH and acidity of the products obtained was observed. A nutritionally favourable increase in the amount of PUFAs was observed, but the 2% oil addition had a slight effect on the deterioration of sensory characteristics [19]. Additionally, and this was not investigated, the introduction of vegetable oil and its refrigerated storage can lead to unfavourable fat quality changes (hydrolysis, oxidation, rancidity). When foods are enriched with non-encapsulated oils (in free form), PUFAs are in direct contact with pro-oxidants. The protective layer acts as a barrier between reactive oxygen species and the oil droplet [20]. Therefore, it may be justified to carry out a nanoencapsulation process of the vegetable oil before incorporating it into the product. This will protect the oil from external factors and improve sensory properties. Similar studies have already been conducted on the introduction of gel-coated encapsulated safflower oil into burger meat (using sodium alginate and konjac flour) [21]. The results obtained contributed to an experiment involving the introduction of nanocapsuled walnut oil in a protein coating into yoghurts. The work of fortifying yoghurts with bioactive ingredients is well known. A new trend is the use of nanocapsules to enrich dairy products [22], and in particular yoghurt [23,24,25]. The encapsulation process involves encapsulating or coating the active substance (core) by a carrier material (wall material, coating, outer phase, membrane) to form capsules or particles on a micrometre or nanometre scale. During the encapsulation process, it is possible to obtain macro- (>1000 μm), micro- (1–1000 μm), and nanocapsules (<1 μm), which have a major impact on improving product functionality [26]. The right nanoencapsulation technique can help protect sensitive compounds from environmental stress factors as well as those during the manufacture of dairy products. This also prevents adverse interactions of the bioactive substances with the compounds present in the milk and improves the organoleptic qualities.

The aim of the current study was to produce nano/microcapsules containing walnut oil, and to use the obtained capsules for the fortification of yoghurt with PUFAs. Only natural ingredients used in yoghurt production were used for encapsulation. Walnut oil was emulsified with milk, and milk powder was used to stabilize the emulsion as a source of protein and lipids. The presence of nano/microcapsules was detected by scanning electron microscopy SEM. Analysis of the yoghurt’s composition, physicochemical properties, microflora, rheological and textural parameters, sensory evaluation, and fatty acid profile were conducted on days 1 and 14 of refrigerated storage.

## 2. Materials and Methods

### 2.1. Materials

Yoghurt was made in laboratory conditions from micro filtered and pasteurized 2% cow milk (District Cooperative Dairy in Piątnica, Piątnica, Poland), skimmed milk powder (Dairy Cooperative in Gostyn Gostyń, Poland), and with Yo-Flex starter culture (Danisco Biolacta, Olsztyn, Poland). Nanoemulsion of walnut oil was made from cold-pressed walnut oil (Oleofarm, Wrocław, Poland) and skimmed milk powder (Dairy Cooperative in Gostyn, Gostyń, Poland).

### 2.2. Nanoemulsion of Walnut Oil Preparation

To create the nanoemulsion, 77.7 mL (80 g) of milk and 80 g of walnut oil were mixed with an ultrasonic homogeniser (20 kHz, Sonopuls HD 4200, Bandelin, Berlin, Germany) until a homogeneous emulsion formed, which took 10 min. Afterward, 20 g of skimmed milk powder was added to stabilise the nanoemulsion. Finally, the resulting nanoemulsion was used to manufacture yogurt.

### 2.3. Yoghurt Preparation

The yogurt was obtained according to the Estrada et al. [3] with modifications. To increase the dry matter content in milk, 2% of skimmed milk powder was added. After that, milk was pasteurized using the tank method for 85 ± 1 °C/20 min, cooled down to 60 ± 1 °C, and divided into two parts. One part was left as a control yoghurt (without oil addition), and an appropriate amount of nanocapsulated walnut oil was added to the second part. Samples (with and without nanocapsules) were then homogenised 3 times in an Armfield homogeniser Ltd. Ringwood (Ringwood, UK) at a pressure of 7 MPa. All the batches were cooled down to the incubation temperature (37 °C) and inoculated. After being poured into a glass container (100 mL), the yoghurts were incubated (8 h/37 °C) in an incubator (CLW 115 ECO Pol-Eco Aparatura, Wodzisław Śląski, Poland) until the pH was 4.6 and stored in a laboratory refrigerator at 4 ± 1 °C for 14 days. For FTIR-ATR spectra and electron microscopy, yoghurt samples were freeze-dried.

### 2.4. Chemical and Physiochemical Properties of Yoghurt Samples

The protein, dry matter, and non-fat dry matter content were analysed in accordance with the procedure [27]. Fat content was measured using the Soxhlet method proceeded by acid hydrolysis [28]. Titratable acidity and pH were measured in accordance with the AOAC official method [27].

### 2.5. Number of Characteristic Microorganisms

Enumeration of the *Lactobacillus delbrueckii* ssp. *bulgaricus* and *Streptococcus thermophillus* [29] found in the yoghurt samples were performed at 1 and 14 days of storage by incubation of the yoghurt sample in MRS and M17 LAB-Agar (BioCorp, Warsaw, Poland) at 37 °C for 48 h and 37 °C for 72 h, respectively. Buffered peptone water (BioCorp, Poland) was used for suspension and dilutions.

### 2.6. Rheological Measurements

Rheological measurements were determined according to Khachatryan et al. [30] with some modification, using a RheoStress RS 6000 (Thermo Scientific, Karlsruhe, Germany) rotary rheometer equipped with a plate–plate P 35 Ti geometry. The temperature of the base plate was 25.0 ± 0.1 °C. The measurement was carried out on freshly prepared samples (1 day) and after 14 days of storage in the fridge at 8 °C. On the measurement day, the sample was removed from the refrigerator and incubated at 25 °C for 1 h. The measurements were run in triplicate.

*Flow curves:* The shear rate was raised from 0.1 to 300 s^−1^ over a 10 min period and a subsequent decrease of shear rate from 300 to 0.1 s^−1^ over a 10 min period. Obtained flow curves were described by Ostwald de-Waele rheological model:$$\tau =K\cdot {\dot{\gamma }}^{n}$$
where: *τ*—shear stress (Pa), *K*—consistency coefficient (Pa·s^n^), $\dot{\gamma }$—shear rate (s^−1^), and *n*—flow behaviour index.

*Oscillation stress sweep test:* the stress was increased from 0 to 300 Pa in 40 logarithmic steps at constant frequency (1 Hz).

*Frequency sweep test:* frequency was increased from 0.01 to 30 Hz at 1Pa deformation fitting the range of linear viscoelasticity.

### 2.7. Texture Analysis

A texture analyser (TA.XTplus, Stable Micro Systems Ltd., Godalming, UK) was used to analyse the textural properties of yoghurt samples. A TPA test was performed. The disc plunger with a diameter of 20 mm at a speed of 1 mm/s^−1^, with a force F = 10 g plunge twice in a depth of 25 mm, into the sample container (55 mm diameter jar). The temperature of the test samples was 4 ± 1 °C. As a result of the test, graphs were obtained showing the dependence of the force on the distance travelled by the probe. Based on the graphs, the hardness (G/N), adhesiveness (G·s/N·s), cohesiveness (-), and chewiness (G/N) were determined.

### 2.8. Scanning Electron Microscope (SEM) Analysis

The freeze-dried yoghurt sample containing the nanocapsules underwent analysis of its nanoparticle size and morphology by means of a JEOL 7550 scanning electron microscope (Akishima, Tokyo, Japan). Prior to the measurements, the sample was sprayed (K575X Turbo Sputter Coater, Emitech, Ltd., Kent, UK) with 20 nm Chromium (Cr) to increase the conductivity of the sample.

### 2.9. ATR-FTIR Spectroscopy

The ATR-FTIR spectra of the freeze-dried yogurt samples were analysed in a wavelength range of 4000–700 cm^−1^ using a MATTSON 3000 FT-IR spectrophotometer (Madison, WI, USA), equipped with a 30SPEC 30 Degree Reflectance accessory (MIRacle ATR, PIKE Technologies Inc., Madison, WI, USA). Measurements were made at a temperature of 25 °C (±2 °C).

### 2.10. Sensory Analysis

A 5-point scale (from 1—bad quality to 5—very good quality) [31] was used for all yoghurt samples. Weighting factors were established for individual discriminants. A trained team of 14 people conducted the evaluation in in three separate series (representing three production series of each type of fermented milk) under appropriate analysis conditions (sample preparation, experimental conditions). The analysis was based on comparing the quality of individual characteristics of the assessed sample with the definitions listed in the evaluation sheet and the score assigned to a given definition into the card. Each member of the team assessed the samples individually. The particular scores’ characterizing feature, multiplied by the corresponding weighting factor, made up the overall rate. In addition, taste, aroma, consistency, and appearance were assessed on a 9-point hedonic scale. Point 1 meant disliked extremely and point 9—liked extremely. The conditions of the analysis were the same as for the evaluation in 5-point scale.

### 2.11. Fatty Acid Analysis

Chloroform and methanol were used in a 2:1 volume ratio to extract the yoghurt sample [32]. The extracted fat was then esterified using the proposed method [31]. A Thermo Electron Corporation TRACE GC ULTRA chromatograph equipped with a flame ionisation detector (FID) and a BPX-70 chromatographic column (60 m × 0.20 mm) with a stationary phase thickness of 0.25 m was used for the chromatographic analysis of the samples. Analyses were carried out using the temperature program with the following temperature settings: initial temperature of 60 °C and hold at this temperature for three minutes; temperature increase of 7 °C per minute until the temperature reaches 200 °C and hold at this temperature for twenty minutes. Dispenser and detector temperatures are 220 °C and 10:1 fractional dosing, respectively. The carrier gas is helium (5 mL/min). The individual peaks were identified by comparing the retention times of each peak with those of the standard fatty acid methyl ester (Supelco 37 FAME Mix, Sigma-Aldrich Co., St. Louis, MO, USA) and CLA isomers (Sigma-Aldrich Co., St. Louis, MO, USA). The concentration of each fatty acid was calculated from the internal standard peak area and expressed as milligrams per 100 g of product. Each sample was analysed in duplicate. Two injections were made into the GC per duplicate (*n* = 2 × 2 × 3). The thrombogenic (TI) and atherogenic (AI) indices were calculated according to [33].

### 2.12. Statistical Analysis

The experiments were carried out in 3 independent series. In turn, each analysis was carried out in duplicate on the 1st and 14th day of refrigerated storage. The results were statistically evaluated using Statistica 13.3 software (TIBCO Software Inc., Palo Alto, CA, USA). Means and standard deviations were calculated. Statistical analysis of the basic chemical composition was performed via one-way ANOVA, in which the type of yoghurt (with or without oil) was a factor. For other results, two-way ANOVA was carried out (factors: type of yoghurt and duration of refrigerated storage). The Duncan’s test at the significance level: *p* ≤ 0.05 was performed to show the significance of differences between the mean values. The information “not stated” (ns), indicates that there was no statistically significant effect of the factors on the values given in the table.

## 3. Results and Discussion

### 3.1. Chemical and Physiochemical Properties

Table 1 shows the results for compositional analysis of the yoghurts. The results obtained for the determined components are in accordance with the guidelines of the Codex Alimentarius Standard [34]. One-way analysis of variance showed the effect of oil addition on fat and dry matter content. A higher amount of these components was found in yoghurt with oil addition, which is due to the additive used. An increase in fat and dry matter was also found in kefir with oil addition as described in Turek and Wszołek [19].

The parameters of potential and active acidity of yoghurts, reflecting the course of the fermentation process, are presented in Table 2. The values obtained testify to the correct course of lactic fermentation, the appropriate acidifying activity of the microorganisms used, their viability, and the appropriate parameters of milk incubation. These values, are also in accordance with PN-A-86061:2002 [35]. Statistical analysis of the results obtained showed the effect of storage time on titratable acidity and pH. In addition, the effect of oil addition on the pH of the yoghurts was found. As a result of storage, the titratable acidity of the yoghurts increased, and the pH decreased in both yoghurts with oil addition and natural yoghurts. It indicates an ongoing fermentation process even under refrigeration conditions. As reported by Mani-Lopez et al. [36], the changes in pH and acidity are a result of the microorganisms used in the yoghurt production process. The lower pH of fresh yoghurts with added oil may be due to the oil presence or a larger amount of acidifying culture in this product. Baba et al. [4] found a lower pH of yoghurts fortified with linseed and walnut oil emulsion in water, explaining this fact by the inhibitory effect of this additive. Ilyasoglu and Yilmaz [8] observed a significant increase in acidity and a decrease in the pH parameter of yoghurts fortified with walnut pulp, explaining the results by the abundance of the characteristic microflora, which was less abundant in the control yoghurt.

### 3.2. Microbial Analysis

Microbiological analysis of yoghurts was performed on the first and fourteenth days of storage (Table 3). In the yoghurts produced, the predominant microbial species was the lactic streptococcus *S. thermophilus*, and their abundance, at 109 cfu/mL, was in line with the reports of Mani-Lopez et al. [36]. Statistical analysis showed no effect of the addition of nanoencapsulated walnut oil on the abundance of labelled microorganisms in the yoghurts tested. However, the effect of storage time on the abundance of *Lactobacillus delbrueckii* ssp. *bulgaricus* was shown. After 14 days, an increase of one logarithmic cycle in the abundance of these microorganisms was observed, indicating the ability of these microorganisms to multiply in the environment with vegetable oil.

The absence of impact from the addition of oil on microbial counts is a favourable occurrence. This fact is indicative of the high stability of the starter bacteria during the shelf life. According to Van Nieuwenhove et al. [37], high amounts of linoleic, oleic, and linolenic acid can have an inhibitory effect on microbial growth. The reported authors also observed a high tolerance of *S. thermophilus* bacteria to high concentrations of linoleic acid in buffalo milk cheeses with sunflower oil. In contrast to their own study, in the experiment conducted by Baba et al. [4], a storage-dependent reduction in lactic acid bacilli and streptococci was observed in yoghurts enriched with flaxseed oil microemulsion. However, the addition of linseed oil led to a greater decrease in streptococci compared to lactic acid bacilli. Similar results were also obtained by Yangilar and Yildiz [38], who reported an increase in the number of bacilli and a decrease in the number of streptococci in ginger oil yoghurt following refrigerated storage.

### 3.3. Rheology

Figure 1 shows the changes in the flow curves of yoghurts stored for 14 days. Table 4 presents the parameters of the Ostwald de-Waele model fitted to the obtained curves. The values of apparent viscosity at a given shear rate are listed in Table 5. The yoghurts studied in this paper exhibit a significant deviation from Newtonian fluids. The parameter *n* indicates that these are pseudoplastic fluids, diluted by shear (*n* < 1). The shear changes the effective volume of the sample, molecules untangle due to orientation in flow direction, and the viscosity decreases [39,40]; certain intermolecular bonds governing the structure and integrity of the network may be disrupted, and the gel is reorganized [41].

The apparent viscosity values of yoghurt with the addition of nanoparticles (sample O) are slightly lower than the plain yoghurt–sample N (Table 5). However, as confirmed by sensory analysis, these differences do not affect product characteristics such as taste, aroma, and consistency (Table 5). It is extremely important information, because as reported in the literature, an increase of oil concentration in emulsion resulted in an increase in the *K* value, and the flow index was altered to a high degree of pseudoplasticity [42]. Thus, the form of introduction of oil into the emulsion is also important both from the point of view of its properties and sensory characteristics of the product. The applied fortification procedure, on the one hand, allowed the introduction of an additional amount of oil with more favourable parameters, and on the other hand, thanks to the nano/microencapsulation technique, the rheological and sensory parameters remained at a similar level to the control sample (sample N). This is valuable knowledge for enriching yoghurts with health-promoting encapsulated oils. The direct addition of walnut oils to yoghurt could be limited due to their impact on the rheological and sensory properties of the product. Only the use of the encapsulation method offers the chance to introduce fats into the product without a clear and negative impact on the quality parameters in question.

Studying the changes occurring during storage of the yoghurts, a gradual decrease in shear stress with time was noted in both cases, indicating a decrease in the apparent viscosity of the systems (Table 5). The emulsions under study showed significant thixotropy, as evidenced by hysteresis in Figure 1. The thixotropic behaviour of an emulsion demonstrates the existence of a weak structure that breaks down under shear. After the shear ceases, the structure partially or fully returns to its original state [43].

The strain sweep behaviour of the storage and loss moduli of the yoghurts were presented in the Figure 2. The analysis was carried out to obtain the linear viscoelastic region (LVR) and gives valuable information on the structural stability of yoghurt variants [39]. The maximum values of strain at which the magnitudes of the moduli G′ and G″ are constant determine the range of linear viscoelasticity. The size of the range of linear viscoelasticity indicates the stability of materials. The larger the range of this parameter, the greater the stability and strength of the material. The G′ (storage modulus) characterized the elastic behaviour of material, while the G″ (loss modulus) represents the viscous nature of the sample. The yoghurts showed comparable behaviour, regardless of whether they were fresh or stored for 14 days. A plateau region was observed in which the values of mechanical moduli were constant. The plateau area is ascribed to the formation of physical connections between biopolymer molecules, resulting in a complex 3D network [44]. In the case of yoghurt, whey proteins are denatured during production, and consequently their covalent bonding with casein micelles is considered one of the conditions for a strong yoghurt structure [45]. Once the plateau region was reached, the mechanical response of the sample showed a sharp decrease for high stresses, corresponding to a mechanical disruption of the molecular arrangement of the yoghurt [39]. The storage modulus exceeds loss moduli (G′ > G″) for values of small deformations, which confirms the semi-solid nature–gel yoghurt structure that develops during fermentation [39,42,46,47]. With the passage of time, the stability of the tested yoghurts remained at a high level. Outside the range of linear viscoelasticity, irreversible changes occurred in the structure of the yoghurts due to the applied strain (Figure 2).

The frequency sweep of the yoghurts is presented in Figure 3. The storage modulus (G′) was higher than the loss modulus (G″) over the entire frequency range, revealing that the elastic deformations dominated over the viscous dissipation mechanisms, and exposes a solid-like behaviour of the yoghurts structure, which was also confirmed by Trujillo-Ramirez et al. [39].

The high stability of the fortified sample demonstrates the beneficial effect of the additive used. The viscoelastic properties of the tested system reflect the interactions between aggregates during oscillation [48]. In natural yoghurt, the structure is stabilized by milk protein–lipid interactions [39]. Substitution of part of the milk fat contributes to a decrease in elasticity and loss modulus. This was confirmed in an earlier study by Pang et al. [49], in which milk fat was replaced by bovine and fish gelatin, and a study by Trujillo-Ramirez et al. [39], where half of the milk fat was substituted by a canola oil-in-water emulsion. This limited the interaction between the different yoghurt components, so that the mechanical properties were weakened. In the present study, we did not reduce the amount of milk fat, but additionally introduced vegetable oil, so that the amount of dry matter increased (Table 5) and the changes in viscoelastic properties were imperceptible.

### 3.4. Texture

The results for the texture of the yoghurts are shown in Table 6. Statistical analysis showed an effect of vegetable oil addition on the hardness and chewiness of the yoghurts, while the addition had no effect on adhesiveness and cohesiveness. Significantly lower values of these parameters were found for yoghurts with nanoencapsulated walnut oil. The results obtained in our study are a consequence of the presence of unsaturated fatty acids with lower melting points than milk fat. In a study by Goyal et al. [14] on the production of Indian dahi fermented milk with mesophilic microbial culture with the addition of microencapsulated flaxseed oil, the authors also found lower firmness values in yoghurts with oil compared to control yoghurts. Similar results were also obtained by Veena et al. [50] in dahi with the addition of flaxseed oil emulsion in water and Bobe et al. [51].

Refrigerated storage of the yoghurts did not affect all the texture parameters determined, as did the refrigerated storage of kefirs with walnut oil and *Camelina sativa* oil described in Turek and Wszołek [19].

### 3.5. Sensory Analysis

Sensory evaluation, especially taste, aroma, consistency, and appearance, is a key tool enabling assessment of consumer acceptance of a new product. The results of the sensory evaluation of the control yoghurts and the yoghurts enriched with nanoencapsulated walnut oil are presented in Table 7. Statistical analysis demonstrated that refrigerated storage influenced the taste of the yoghurts. The stored yoghurts without added oil received lower taste scores. The opposite trend was observed for yoghurts with added oil, where the products scored better after storage. The oil supplementation was not found to affect the flavour ratings of the yoghurts. In terms of smell and texture, yoghurts without added oil were rated better. In terms of appearance, natural yoghurts also scored better, with all yoghurts scoring lower on this characteristic after 14 days of storage. Presumably, this is related to the occurrence of syneresis, caused by proteolysis of milk proteins through the action of bacterial enzymes. In the overall evaluation, fresh yoghurts without added oil and yoghurts with added oil scored more favourably after refrigerated storage. The opposite trend was observed for yoghurt with 2% addition of unencapsulated walnut oil, where fresh kefirs with oil were rated more favourably and lower after 14 days of storage compared to control kefirs [19].

The absence of a statistically substantial impact of the introduction of vegetable oil on the flavour of yoghurts is a positive outcome, implying the suitability of this commodity. Goyal et al. [16] also reported no effect of the addition of microencapsulated linseed oil at 2% on the taste of Indian dahi yoghurt. The other distinguishing characteristics of the sensory evaluation of the yoghurts with added oil were rated slightly worse by the panellists than the control yoghurts. The sensory quality analysis conducted by Ammar et al. [52] for yoghurts with different levels of olive oil addition to probiotic yoghurts showed that the addition improved the ratings for texture, consistency, and flavour of fresh products, while after storage, these quality attributes were rated slightly worse. In contrast, a study by Barrantes et al. [53] indicated that the addition of vegetable oils to yoghurt negatively affected the sensory quality evaluation of these products. It was noted that this effect could be minimised by the addition of sugar or flavourings. The slightly lower acceptability of yoghurts with added vegetable oils in the study by Dal Bello et al. [12] and the slight negative effect of the addition of nanoencapsulated walnut oil in our own study demonstrate the possibility of obtaining a consumer-acceptable, functional fermented milk with increased unsaturated fatty acid content. In addition, the more favourable overall evaluation scores obtained after storage of yoghurts with the addition of oil subjected to the nanocapsule compared to stored kefirs with the addition of oil without the nanocapsule [19] may attest to the effective masking of the negative organoleptic characteristics of the oil caused by fat decomposition.

### 3.6. Scanning Electron Microskopy (SEM)

The SEM analysis illustrated in Figure 4 of the freeze-dried yoghurt samples showed that nano/micro spherical structures were present. These structures arose from the encapsulation of oil and milk emulsions within proteins and lipids present in powdered milk, leading to the creation of multilayer shells. Uniformly distributed nano/microstructures, ranging in size from several hundred nanometres to several micrometres, are dispersed throughout the matrix.

During the process of freeze-drying and when capturing images under an electron microscope, the capsules obtained were deformed and opened due to the influence of vacuum and high energy from electron bombardment. However, this allowed for the internal structure of the capsules to be scanned, and the formation of micellar structures was observed. The photos also display intact capsules of varying sizes ranging from several hundred to 1000 nm. The micelles obtained exhibit multidimensionality due to their complex structures, suggesting the formation of nano/micro capsules.

### 3.7. FTIR-ATR

FTIR spectroscopy was used to monitor changes in the structure of the yoghurt before and after fortification. The results of the FTIR spectra of yoghurt, yoghurt containing walnut oil, and walnut oil are shown in Figure 5. In both spectra, the main characteristic bands associated with the proteins, carbohydrates, and fats present in the yoghurt samples were observed. In both yoghurt samples, a broad band is observed in the range from approximately 3000 to 3500 cm^−1^, which relates to the vibration of the hydroxyl group. In both samples, bands were observed at 2920 cm^−1^ (symmetric stretching vibration of CH_2_-groups of fatty acids) and at 2854 cm^−1^ (asymmetric stretching vibration of CH_2_-groups in membrane lipids) [54,55]. We also observe characteristic bands for the first- and second-order amide groups at 1630 and 1530 cm^−1^, respectively.

We do not observe significant differences between the spectra of the yoghurts. The most significant differences are due to the presence of oil; we observe an increase in the intensity of the bands at 2920 and 2854 cm^−1^ (methylene groups), 1623 and 1744 cm^−1^ C=C and C=O ester carboxylic acids, and 1457 and 1377 cm^−1^ (methylene and methyl groups bending vibrations).

### 3.8. Fatty Acids

Table 8 shows the fatty acid profile of the analysed yoghurts. Statistical analysis showed an effect of the addition of vegetable oil on most of the fatty acids analysed. The exceptions were stearic acid (C18:0) and vaccenic acid (trans-11 C18:1), the proportion of which did not change due to the addition of oil to the yoghurts and their storage. This is not consistent with reports by Turek and Wszołek [56], where a statistically significant reduction in these fatty acids was found in probiotic kefirs with 0.03% added walnut oil produced with two types of starter cultures, due to a decrease in bacterial lipase activity associated with a reduction in kefir microbiota during storage. In the yoghurts tested, the microflora was active and increased during storage, so the proportion of these fatty acids did not change. In addition, there may have been a reduction in the bacterial lipase activity of the yoghurt culture as a result of the nanocapsule oil.

Refrigerated storage of the yoghurts only affected the proportion of decenoic acid (C10:1) and pentadecenoic acid (C15:1). A significantly lower contribution of short-chain fatty acids was observed in yoghurt with added oil. Butyric (C4:0), caproic (C6:0), caprylic (C8:0), capric (C10:0), and lauric (C12:0) acids are fatty acids characteristic of milk fat, so their proportion in natural yoghurt was higher. The proportion of medium-chain fatty acids was also lower, with the exception of decenoic acid (C10:1), which was higher in yoghurt with oil and increased as a result of storage. The percentage of C15:1 acid also increased as a result of storage in both types of yoghurt. In the probiotic kefirs with 0.03% added walnut oil produced by Turek and Wszołek [56], a statistically significant increase in decenoic acid and other monounsaturated fatty acids was also observed as a result of storage, confirming a different bioconversion mechanism for these fatty acids or their non-use as substrates in the enzymatic processes carried out by the yoghurt culture microorganisms.

Milk fat contains significant amounts of saturated fatty acids such as myristic acid, stearic acid, and palmitic acid [57]. Walnut oil is a source of unsaturated acids—oleic, linoleic, and linolenic acids [58]. Replacing saturated fatty acids with unsaturated ones may serve to prevent the occurrence of lifestyle diseases, mainly cardiovascular diseases [3]. Long-chain fatty acids, mainly linoleic acid, α-linolenic acid, and arachidic acid, increased as a result of the addition of oil. Analogously to kefirs with the addition of two levels of walnut oil and groundnut oil [19] and probiotic kefirs with the addition of walnut oil [56], the increase in the proportion of long-chain fatty acids is due to the addition of vegetable oil, which is richer in these fatty acids. This results in a decrease in the proportion of saturated and monounsaturated acids while increasing polyunsaturated acids. The 2% addition of nanocasted walnut oil to yoghurt resulted in a more than 10-fold increase in the proportion of PUFA acids, and an almost 2-fold reduction in the proportion of SFA acids. From a nutritional point of view, the reduction in the proportion of saturated and the increase in polyunsaturated fatty acids is beneficial. Fat quality indices, in the form of TI (thrombogenic) and AI (atherogenic) indices, are considered better indicators of the dietary and health value of fat than the PUFA/SFA ratio. It is assumed that the lower their value, the more favourable the fatty acid profile in terms of health. Yoghurts with nanoencapsulated vegetable oil showed much more favourable values for these ratios than natural yoghurts. Similar values of the aforementioned indices were obtained in kefir with a 2% addition of walnut oil [19].

## 4. Conclusions

In the work, it was possible to successfully enclose walnut oil in the form of nano/microcapsules and use it in food fortification. Such a treatment allowed for the enrichment of yoghurt with PUFAs, ingredients involved in many metabolic and physiological processes of the body. The analysis confirmed that the fortified yoghurts had an increased content of fat, dry matter, and higher pH than the control product, which resulted from the addition of walnut oil in the encapsulated form. The share of most of the analysed fatty acids changed due to yoghurt fortification, which was not observed after 14 days of refrigerated storage. In the dietary terms, yoghurts with the addition of oil show a more favourable share of the examined groups of fatty acids (SFA, MUFA, and PUFA) and lower values of the analysed ones’ fat quality indices (AI and TI). Based on the microbiological analysis, no differences in the number of microorganisms in the tested yoghurts were found. However, the impact of storage time (14 days) on the number of *Lactobacillus delbrueckii* ssp. *bulgaricus* bacteria was observed. Textural and rheological analyses showed slight differences between the samples, but the observed changes did not significantly affect the sensory attributes of the product. The fortified yoghurts were highly appreciated by the panellists, and the 14-day storage period did not affect the deterioration of sensory characteristics. Based on the obtained results, it can be concluded that the enrichment of yoghurt with encapsulated walnut oil has a positive effect on its physicochemical and functional properties. Achieving nanoemulsions of oil in skimmed milk powder, which is commonly used in yoghurt production, offers great potential for the use of this ingredient in the production of other dairy products such as rennet cheese, quark, or ice cream. An attempt should also be made to clarify any reactions and mechanisms taking place between the oil nanocapsules and the food matrix.

## Figures and Tables

**Figure 1 foods-12-03842-f001:**
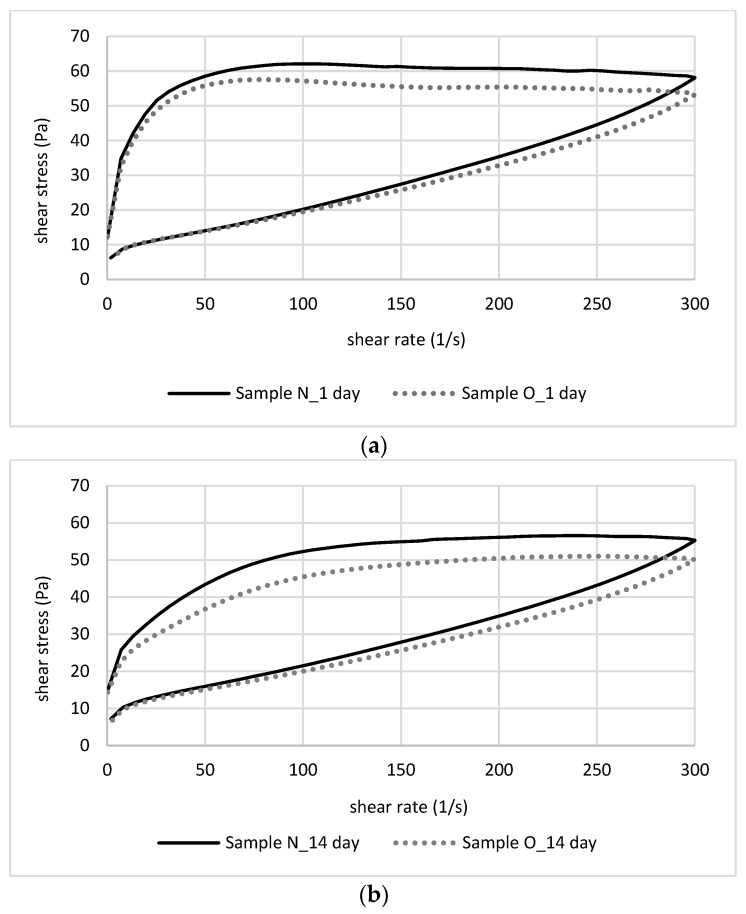
Flow curves of yoghurts tested at (**a**) 1 and (**b**) 14 days of storage.

**Figure 2 foods-12-03842-f002:**
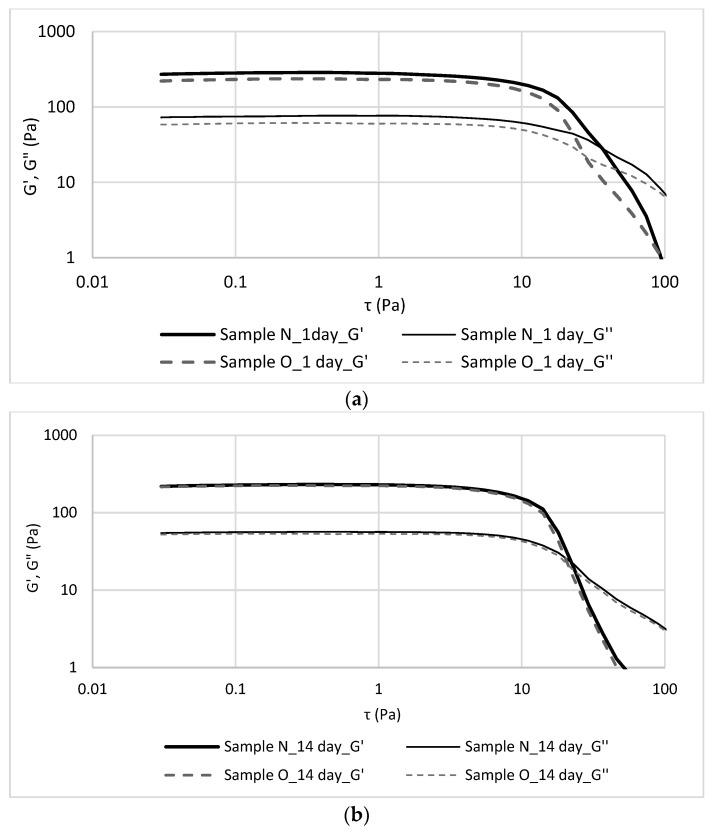
Values of storage and loss moduli (G′ and G″) of sample N O, measured in stress sweep test at (**a**) 1 and (**b**) 14 days of storage.

**Figure 3 foods-12-03842-f003:**
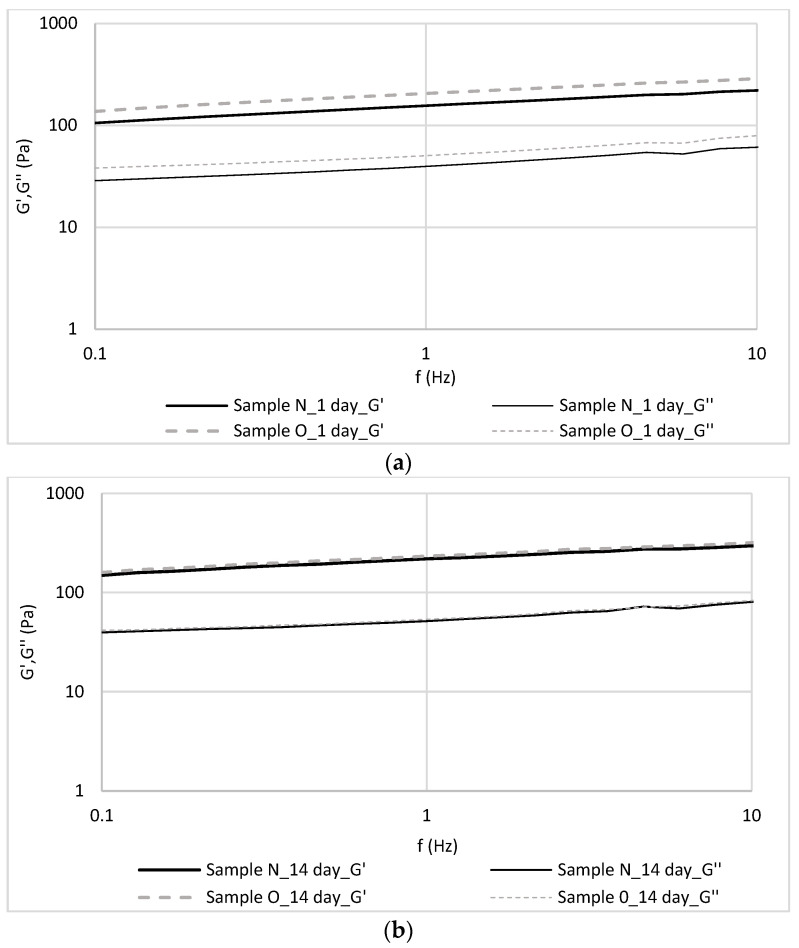
Values of storage and loss moduli (G′ and G″) of yoghurts at (**a**) 1 day of storage and (**b**) 14 days of storage.

**Figure 4 foods-12-03842-f004:**
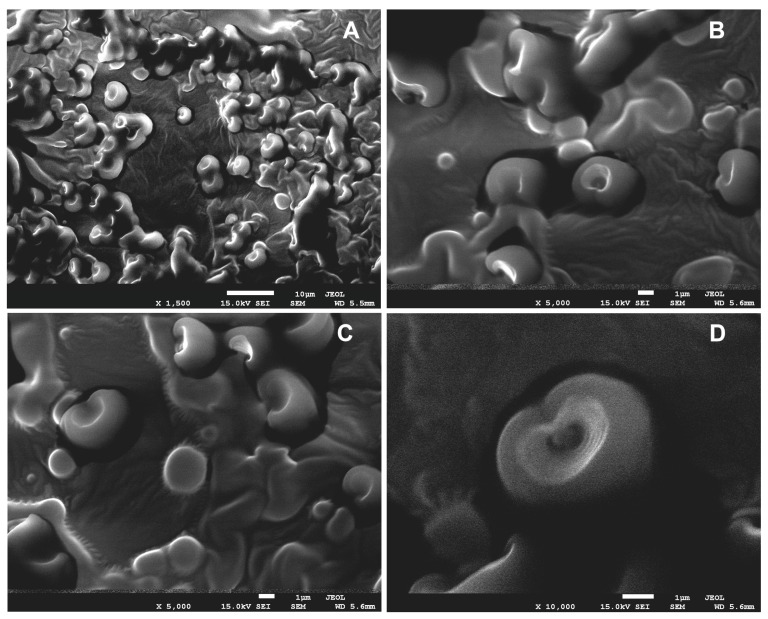
SEM images of a yoghurt sample fortified with nano-/microcapsules containing walnut oil at magnifications of ×1500 (**A**), ×5000 (**B**,**C**), and ×10,000 (**D**).

**Figure 5 foods-12-03842-f005:**
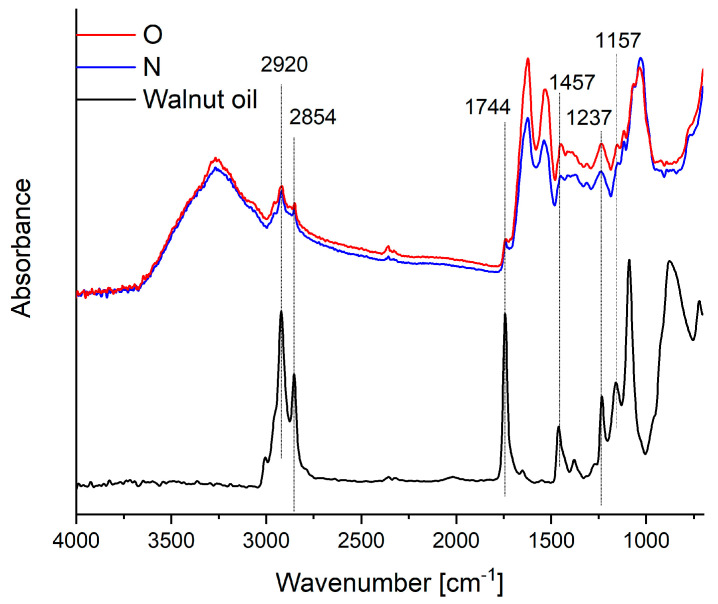
FTIR-ATR spectra of walnut oil, yoghurt (N), and yoghurt with the addition of nanoencapsulated walnut oil (O).

**Table 1 foods-12-03842-t001:** Chemical composition of yoghurt with the addition of nanoencapsulated walnut oil (mean ± SD).

	Yoghurt Type	
	N	O
Protein [%]	4.08 ^a^ ± 0.04	4.02 ^a^ ± 0.02
Fat [%]	2.01 ^a^ ± 0.06	4.04 ^b^ ± 0.04
Dry matter [%]	10.78 ^a^ ± 0.47	12.62 ^b^ ± 0.03

N—yoghurt without oil addition. O—yoghurt with 2% of walnut oil nanoemulsion addition. ^a–b^ different letters in the same row indicate significant differences (ANOVA, *p* ≤ 0.05).

**Table 2 foods-12-03842-t002:** Physicochemical parameters of yoghurt with the addition of nanoencapsulated walnut oil on the first day after production and after 14 days of refrigerated storage (mean ± SD).

Parameters	Storage Time (Days)	Yoghurt Type		Factor Effects	
		N	O	Yoghurt Type Storage Time	
Titrable acidity [% of lactic acid]	1	1.03 ^a^ ± 0.02	0.9 ^a^ ± 0.04	ns	0.004
	14	1.13 ^b^ ± 0.01	1.13 ^b^ ± 0.02		
pH parameters	1	4.35 ^Aa^ ± 0.00	4.26 ^Ba^ ± 0.00	0.002	<0.001
	14	4.21 ^Ab^ ± 0.01	4.21 ^Bb^ ± 0.00		

N—yoghurt without oil addition. O—yoghurt with 2% of walnut oil nanoemulsion addition. ns—not stated. ^a–b, A–B^—different superscript lowercase letters in the same column and different uppercase letters in the same row indicate significant differences (two-way ANOVA, Duncan test. *p* ≤ 0.05) with respect to storage time and oil addition, respectively.

**Table 3 foods-12-03842-t003:** Number of microorganisms in yoghurt with the addition of nanoencapsulated walnut oil during storage [log CFU/mL] (mean ± SD).

Microorganisms Type	Storage Time (Days)	Yoghurt Type		Factor Effects	
		N	O	Yoghurt Type Storage Time	
*Lactobacillus delbrueckii* ssp. *bulgaricus*	1	3.64 ^a^ ± 0.01	3.59 ^a^ ± 0.01	ns	<0.001
	14	4.38 ^b^ ± 0.3	4.63 ^b^ ± 0.15		
*Streptococcus thermophillus*	1	9.58 ^a^ ± 0.01	9.69 ^a^ ± 0.01	ns	ns
	14	9.50 ^a^ ± 0.10	9.54 ^a^ ± 0.10		

N—yoghurt without oil addition. O—yoghurt with 2% of walnut oil nanoemulsion addition. ns—not stated. ^a–b^—different letters in the same column indicate significant differences (two-way ANOVA, Duncan test. *p* ≤ 0.05) for each group of microorganisms.

**Table 4 foods-12-03842-t004:** The parameters of Ostwald de-Waele rheological model and area of hysteresis.

Parameters		Storage Time (Days)	Yoghurt Type		Factor Effects	
			N	O	Yoghurt Type	Storage Time
Ostwald de-Waele model	*K* (Pa/s^n^)	1	35.79 ^a^ ± 1.40	34.74 ^a^ ±0.40	ns	ns
		14	21.40 ^a^ ± 0.49	17.20 ^b^ ± 0.84		
	*n* (-)	1	0.10 ^a^ ± 0.01	0.09 ^a^ ± 0.00	ns	ns
		14	0.18 ^b^ ± 0.00	0.20 ^a^ ± 0.01		
	R^2^	1	0.8380	0.7961	-	-
		14	0.9644	0.9751		
Area of hysteresis (Pa/s)		1	8841.00 ^a^ ± 131.52	8022.50 ^b^ ± 4.50	<0.001	<0.001
		14	6482.00 ^a^ ±147.07	5438.67 ^b^ ± 152.95		

*K*—consistency coefficient, *n*—flow behaviour index, R^2^—the coefficient of determination, N—yoghurt without oil addition, O—yoghurt with 2% of walnut oil nanoemulsion addition. ns—not stated. ^a–b^—different letters in the same row indicate significant differences (two-way ANOVA, Duncan test. *p* ≤ 0.05) for each parameter.

**Table 5 foods-12-03842-t005:** Apparent viscosity of yoghurts at increasing shear rate.

Parameters		Storage Time (Days)	Yoghurt Type		Factor Effects	
			N	O	Yoghurt Type	Storage Time
Apparent viscosity (Pa·s)	50 s^−1^	1	1.17 ^a^ ± 0.02	1.12 ^b^ ± 0.01	<0.001	<0.001
		14	0.87 ^a^ ± 0.02	0.74 ^b^ ± 0.02		
	100 s^−1^	1	0.63 ^a^ ± 0.01	0.58 ^b^ ± 0.01	<0.001	<0.001
		14	0.53 ^a^ ± 0.01	0.46 ^b^ ± 0.01		
	200 s^−1^	1	0.31 ^a^ ± 0.00	0.28 ^b^ ± 0.00	<0.001	<0.001
		14	0.28 ^a^ ± 0.01	0.26 ^b^ ± 0.00		
	300 s^−1^	1	0.19 ^a^ ± 0.00	0.18 ^b^ ± 0.00	<0.001	<0.001
		14	0.18 ^a^ ± 0.00	0.17 ^b^ ± 0.00		

N—yoghurt without oil addition, O—yoghurt with 2% of walnut oil nanoemulsion addition.. ^a–b^—different letters in the same row indicate significant differences (two-way ANOVA, Duncan test. *p* ≤ 0.05) for each parameter.

**Table 6 foods-12-03842-t006:** Texture parameters of yoghurt with the addition of nanoencapsulated walnut in the first and last day of refrigerated storage (means ± SD).

Texture Parameter	Storage Time (Days)	Yoghurt Type		Factor Effects	
		N	O	Yoghurt Type	Storage Time
Hardness [N]	1	1.20 ^a^ ± 0.03	1.05 ^b^ ± 0.01	<0.001	ns
	14	1.24 ^a^ ± 0.02	0.99 ^b^ ± 0.006		
Adhesiveness [N·s]	1	3.20 ^a^ ± 0.18	3.56 ^a^ ± 0.18	ns	ns
	14	3.50 ^a^ ± 0.44	3.17 ^a^ ± 0.23		
Cohesiveness [-]	1	0.44 ^a^ ± 0.00	0.45 ^a^ ± 0.01	ns	ns
	14	0.43 ^a^ ± 0.01	0.44 ^a^ ± 0.00		
Chewiness [N]	1	0.50 ^a^ ± 0.02	0.45 ^b^ ± 0.00	<0.001	ns
	14	0.49 ^a^ ± 0.01	0.42 ^b^ ± 0.01		

N—yoghurt without oil addition. O—yoghurt with 2% of walnut oil nanoemulsion addition. ns—not stated. ^a–b^—different letters in the same row indicate significant differences (two-way ANOVA, Duncan test. *p* ≤ 0.05) for each texture parameters.

**Table 7 foods-12-03842-t007:** Sensory evaluation parameters on a 5-point scale of yoghurt with the addition of nanoencapsulated walnut oil on the first and last day of refrigerated storage (means ± SD).

Quality Factors [Point]	Time (Days)	Yoghurt Type		Factor Effects	
		N	O	Yoghurt Type	Storage Time
Taste	1	4.92 ^a^ ± 0.05	4.25 ^a^ ± 0.00	ns	0.002
	14	3.95 ^b^ ± 0.09	4.70 ^b^ ± 0.06		
Aroma	1	4.86 ^a^ ± 0.02	4.63 ^b^ ± 0.07	<0.001	ns
	14	4.95 ^a^ ± 0.03	4.60 ^b^ ± 0.00		
Consistency	1	4.96 ^a^ ± 0.02	4.67 ^b^ ± 0.00	0.039	ns
	14	4.80 ^a^ ± 0.12	4.80 ^b^ ± 0.00		
Appearance	1	5.00 ^Aa^ ± 0.00	4.63 ^Ba^ ± 0.07	0.029	0.010
	14	4.55 ^Ab^ ± 0.14	4.35 ^Bb^ ± 0.14		
Overall assessment	1	4.94 ^Aa^ ± 0.01	4.50 ^Ba^ ± 0.03	<0.001	<0.001
	14	4.47 ^Ab^ ± 0.02	4.62 ^Bb^ ± 0.06		

N—yoghurt without oil addition. O—yoghurt with 2% of walnut oil nanoemulsion addition ns—not stated. ^a–b, A–B^—different superscript lowercase letters in the same column and different uppercase letters in the same row indicate significant differences (two-way ANOVA, Duncan test. *p* ≤ 0.05) with respect to storage time and oil addition. respectively.

**Table 8 foods-12-03842-t008:** Fatty acids profile of yoghurt with the addition of nanoencapsulated walnut oil the day after production and after 14 days of refrigerated storage (means ± SD).

Fatty Acids (%)	Time (Days)	Yoghurt Type		Factor Effects		
		N	O	Yoghurt Type	Storage Time	Interactions
C4:0	1	1.21 ^a^ ± 0.00	0.50 ^b^ ± 0.06	<0.001	ns	ns
	14	1.06 ^a^ ± 0.52	0.58 ^b^ ± 0.00			
C6:0	1	1.04 ^a^ ± 0.00	0.46 ^b^ ± 0.05	<0.001	ns	ns
	14	0.98 ^a^ ± 0.05	0.50 ^b^ ± 0.00			
C8:0	1	0.80 ^a^ ± 0.00	0.40 ^b^ ± 0.04	<0.001	ns	ns
	14	0.79 ^a^ ± 0.03	0.30 ^b^ ± 0.06			
C10:0	1	2.25 ^a^ ± 0.00	0.95 ^b^ ± 0.09	<0.001	ns	ns
	14	2.18 ^a^ ± 0.05	1.10 ^b^ ± 0.00			
C10:1	1	0.05 ^Aa^ ± 0.00	0.06 ^Ba^ ± 0.02	<0.001	<0.001	<0.001
	14	0.05 ^Ab^ ± 0.00	0.18 ^Bb^ ± 0.00			
C12:0	1	2.98 ^a^ ± 0.00	1.13 ^b^ ± 0.12	<0.001	ns	ns
	14	2.98 ^a^ ± 0.01	1.47 ^b^ ± 0.00			
C13:0	1	0.09 ^a^ ± 0.00	0.06 ^b^ ± 0.02	0.002	ns	ns
	14	0.10 ^a^ ± 0.00	0.05 ^b^ ± 0.00			
C14:0	1	10.88 ^a^ ± 0.00	4.89 ^b^ ± 0.50	<0.001	ns	ns
	14	11.15 ^a^ ± 0.09	5.43 ^b^ ± 0.02			
C14:1	1	1.15 ^a^ ± 0.00	0.52 ^b^ ± 0.05	<0.001	ns	ns
	14	1.18 ^a^ ± 0.00	0.58 ^b^ ± 0.00			
C15:0	1	0.56 ^a^ ± 0.06	1.20 ^b^ ± 0.00	<0.001	ns	ns
	14	0.60 ^a^ ± 0.00	1.27 ^b^ ± 0.01			
C15:1	1	0.31 ^Aa^ ± 0.00	0.15 ^Ba^ ± 0.01	<0.001	0.037	ns
	14	0.36 ^Ab^ ± 0.05	0.22 ^Bb^ ± 0.01			
C16:0	1	37.02 ^a^ ± 0.03	21.27 ^b^ ± 0.52	<0.001	ns	ns
	14	37.06 ^a^ ± 0.22	21.85 ^b^ ± 0.05			
C16:1	1	2.14 ^a^ ± 0.00	1.14 ^b^ ± 0.02	<0.001	ns	ns
	14	2.08 ^a^ ± 0.02	1.14 ^b^ ± 0.00			
C17:0	1	0.76 ^a^ ± 0.00	0.53 ^b^ ± 0.09	<0.001	ns	ns
	14	0.74 ^a^ ± 0.00	0.42 ^b^ ± 0.00			
C17:1	1	0.39 ^a^ ± 0.00	0.15 ^b^ ± 0.04	<0.001	ns	ns
	14	0.39 ^a^ ± 0.00	0.22 ^b^ ± 0.00			
C18:0	1	11.61 ^a^ ± 0.00	10.31 ^a^ ± 2.96	ns	ns	ns
	14	11.21 ^a^ ± 0.14	7.25 ^a^ ± 0.03			
trans-11 C18:1	1	1.36 ^a^ ± 0.01	1.19 ^a^ ± 0.47	ns	ns	ns
	14	1.57 ^a^ ± 0.05	0.63 ^a^ ± 0.12			
C18:1 n-9	1	21.32 ^a^ ± 0.02	19.69 ^b^ ± 1.21	0.018	ns	ns
	14	21.33 ^a^ ± 0.16	19.28 ^b^ ± 0.22			
C18:2 n-6 trans	1	0.23 ^a^ ± 0.00	0.15 ^b^ ± 0.02	<0.001	ns	ns
	14	0.27 ^a^ ± 0.02	0.16 ^b^ ± 0.01			
C18:2 n-6 cis	1	2.05 ^a^ ± 0.01	30.00 ^b^ ± 2.66	<0.001	ns	ns
	14	1.89 ^a^ ± 0.01	31.93 ^b^ ± 0.30			
C18:3 n-6	1	0.18 ^a^ ± 0.00	0.10 ^b^ ± 0.00	<0.001	ns	ns
	14	0.21 ^a^ ± 0.00	0.10 ^b^ ± 0.00			
C18:3 n-3	1	0.38 ^a^ ± 0.00	5.36 ^b^ ± 0.50	<0.001	ns	ns
	14	0.53 ^a^ ± 0.20	5.71 ^b^ ± 0.05			
CLA	1	0.43 ^a^ ± 0.00	0.20 ^b^ ± 0.01	<0.001	ns	ns
	14	0.45 ^a^ ± 0.01	0.21 ^b^ ± 0.00			
C20:0	1	0.37 ^a^ ± 0.00	5.36 ^b^ ± 0.50	<0.001	ns	ns
	14	0.53 ^a^ ± 0.18	5.71 ^b^ ± 0.06			
C20:1	1	0.14 ^a^ ± 0.00	0.07 ^b^ ± 0.02	0.010	ns	ns
	14	0.15 ^a^ ± 0.00	0.11 ^b^ ± 0.02			
SFA	1	70.05 ^a^ ± 0.00	41.32 ^b^ ± 0.01	<0.001	ns	ns
	14	69.71 ^a^ ± 0.00	39.83 ^b^ ± 0.01			
MUFA	1	26.68 ^a^ ± 0.01	22.86 ^b^ ±0.02	0.002	ns	ns
	14	26.93 ^a^ ± 0.02	22.08 ^b^ ± 0.03			
PUFA	1	3.27 ^a^ ± 0.01	35.83 ^b^ ± 0.02	<0.001	ns	ns
	14	3.36 ^a^ ± 0.02	38.09 ^b^ ± 0.01			
AI	1	2.79 ^a^ ± 0.00	0.72 ^b^ ± 0.02	<0.001	ns	ns
	14	2.79 ^a^ ± 0.02	0.75 ^b^ ± 0.02			
TI	1	3.71 ^a^ ± 0.00	0.86 ^b^ ± 0.00	<0.001	ns	ns
	14	3.57 ^a^ ± 0.01	0.78 ^b^ ± 0.02			

N—yoghurt without oil addition. O—yoghurt with 2% of walnut oil nanoemulsion, AI—Atherogenic Index, TI—Thrombogenic Index. ns—not stated. ^a–b, A–B^—different superscript lowercase letters in the same column and different uppercase letters in the same row indicate significant differences (Two-Way ANOVA, Duncan test. *p* ≤ 0.05) with respect to storage time and oil addition, respectively.

## Data Availability

The data presented in this study are available on request from the corresponding author.

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
