# Peer review of "An Innovative Method for the Production of Yoghurt Fortified with Walnut Oil Nanocapsules and Characteristics of Functional Properties in Relation to Conventional Yoghurts"

_foods, 2023, doi:10.3390/foods12203842_

Round 1

Reviewer 1 Report

There are no line numbers in the manuscript. Please add line number for reviewing.

In the whole article, sentences are very large. It is suggested to make these sentences in small sentences for better understanding of the readers. Authors need to make introduction less elaborative and more sharp with clear indication about the gaps in knowledge and the significance of the present study. There are several terminologies which are introduced abruptly and does not fit in the context of this study. Like films, composite films, parameter of potential. The article is too wordy and word count need to be reduced for conveying the right information. Authors have included all the figures from the Rheometer software. It would be better to make the graphs using values for better clarity and clear graphs. The authors have focused less on discussing their results and outcome, instead they have mentioned lot of previous studies mentioning that similar results were obtained for this study.

Page 2:

What do you mean by full health? Please use proper scientific terminology?

Replace animal products with proper scientific term.

“Yoghurts are the most popular and consumed of all fer-mented milks………………….. the granules Streptococcus thermophilus [10].” This paragraph is too basic. I would suggest you to delete it.

Western diet- w should be capital.

a one-week refrigerated- delete a

Introduction is having very big paragraph. Suggested to divide it into smaller paragraphs.

NCDC, Kamal, India- It is Karnal not Kamal. Please correct it.

Page 3:

did not use the addition of- delete the addition of

this fortification- What do you mean by this fortification ?

nutrient content- It is not the nutrient content. This is compositional analysis.

How the compositional analysis is in accordance with Codex guidelines?

Page 4:

What do you mean by parameter of potential?

“Microbiological analysis of yoghurts was performed immediately after inoculation of milk and on the first and fourteenth days of storage (Table 3)”. It has been mentioned in the material and methods that all the analysis were performed on 1st and 14th day of storage. Please don’t repeat this sentence in every section and sub-section.

Authors mention that there is no influence of oil addition on microbial counts. However, while explaining the pH findings, author mentions that lower pH of fresh yoghurts with added oil, may be indicative of the inhibitory effect of this additive. How does authors justify this?

“The yogurt is an O/W emulsion ……………………………. the ratio of casein to whey protein from the natural ratio in milk (4:1) [33–35].” What is the significance of including this paragraph in the discussion?

Page 7:

biopolymer molecules- Which biopolymer molecule is referred here?

Page 8:

14-day aged- use the term stored instead of aged

Page 4 and page 11: Authors have used the term welcome results. What do you mean with this?

Page 11:

What does this statement “while storage influenced lower ratings for these attributes.” means?

“It should be noted that technical term abbreviations will be explained when first used.” What does this statement mean?

Page 16: Materials and methods

Section 3.1: Mention the reference of the methodology used for the preparation of yoghurt.

tank method in 85 ± 1 °C /20 min- It should be tank method for 85 ± 1 °C /20 min.

amount of nanocapsulated- Remove the space before nanocapsulated.

After the oil addition, the milk was homogenized 3 times in Arm-field Ltd. Ringwood (England) homogenizer under a pressure of 7 MPa- What was the reason for homogenizing three times.

jogurt- Use the uniform terminology throughout the manuscript.

based plate- DO you mean base plate ?

tested samples was 4 ±1 °C.- It should be test samples.

The films obtained were analysed in terms of nanoparticle size- Which films are being referred to?

fabricated composite films- What does this mean ?

In the whole article, sentences are very large. It is suggested to make these sentences in small sentences for better understanding of the readers. The authors need to use proper words. 

Author Response

Authors: We thank the Reviewer for valuable comments on the manuscript. All changes have been applied to the text using the change tracking function. We hope that the revision made has improved the manuscript and that its quality has improved significantly.

Reviewer: There are no line numbers in the manuscript. Please add line number for reviewing.

It was added.

In the whole article, sentences are very large. It is suggested to make these sentences in small sentences for better understanding of the readers. Authors need to make introduction less elaborative and more sharp with clear indication about the gaps in knowledge and the significance of the present study.

Thank you for your valuable comments. The manuscript has been rewritten according to the Reviewer's comments.

There are several terminologies which are introduced abruptly and does not fit in the context of this study. Like films, composite films, parameter of potential. The article is too wordy and word count need to be reduced for conveying the right information.

It was corrected according to Reviewer suggestions.

Authors have included all the figures from the Rheometer software. It would be better to make the graphs using values for better clarity and clear graphs.

The files have been changed according to the reviewer's comments. We hope that the quality of the figures is now appropriate.

The authors have focused less on discussing their results and outcome, instead they have mentioned lot of previous studies mentioning that similar results were obtained for this study.

It was corrected.

Page 2:

What do you mean by full health? Please use proper scientific terminology?

It was corrected.

Replace animal products with proper scientific term.

It was corrected.

 “Yoghurts are the most popular and consumed of all fer-mented milks………………….. the granules Streptococcus thermophilus [10].” This paragraph is too basic. I would suggest you to delete it.

It was corrected.

Western diet- w should be capital.

It was corrected.

a one-week refrigerated- delete a

It was corrected.

Introduction is having very big paragraph. Suggested to divide it into smaller paragraphs.

It has been changed.

NCDC, Kamal, India- It is Karnal not Kamal. Please correct it.

It was corrected.

Page 3:

did not use the addition of- delete the addition of

It was corrected.

this fortification- What do you mean by this fortification ?

It has been changed.

nutrient content- It is not the nutrient content. This is compositional analysis.

It was corrected.

How the compositional analysis is in accordance with Codex guidelines?

The sentence has been changed. Codex Alimentarius specifies the composition of fermented milk. In the case of yoghurts, the protein content should be at least 2.7% and the fat content should not exceed 15%.

Page 4:

What do you mean by parameter of potential?

‘Parameter of potential’ has been changed into the ‘potential acidity’.

 “Microbiological analysis of yoghurts was performed immediately after inoculation of milk and on the first and fourteenth days of storage (Table 3)”. It has been mentioned in the material and methods that all the analysis were performed on 1st and 14th day of storage. Please don’t repeat this sentence in every section and sub-section.

It was corrected.

Authors mention that there is no influence of oil addition on microbial counts. However, while explaining the pH findings, author mentions that lower pH of fresh yoghurts with added oil, may be indicative of the inhibitory effect of this additive. How does authors justify this?

Many thanks to the Reviewer for this question. Indeed, the addition of oil was not found to have an inhibitory effect on the microbial growth of yoghurts stored for 14 days. However, there was a decrease in pH in fresh yoghurts with oil. It is not clear to us what the exact reason for this phenomenon was. We suppose that the oil itself containing fatty acids could have been the reason for this and caused the decrease in pH. In addition, as can be seen in Table 3, on day 1 of storage of the yoghurts the number of the strongly acidifying S. thermophilus culture was slightly higher in the yoghurts with oil, which may have influenced the lower pH. After storage, the pH of the yoghurts with and without added oil reached identical values, which may indicate that there were no differences in the acidifying activity of the microorganisms used.

This sentence has been modify.

 “The yogurt is an O/W emulsion ……………………………. the ratio of casein to whey protein from the natural ratio in milk (4:1) [33–35].” What is the significance of including this paragraph in the discussion?

We thank the reviewer for bringing this to our attention. After reconsidering the text, we found that the paragraph was not relevant, it was just an introduction to the discussion. It has been removed.

Page 7:

biopolymer molecules- Which biopolymer molecule is referred here?

It was explained in the text.

Page 8:

14-day aged- use the term stored instead of aged

It was corrected according Reviewer suggestion.

Page 4 and page 11: Authors have used the term welcome results. What do you mean with this?

It was corrected.

Page 11:

What does this statement “while storage influenced lower ratings for these attributes.” means?

The sentence has been modify.

 “It should be noted that technical term abbreviations will be explained when first used.” What does this statement mean?

A phrase from the MDPI form crept in. The text has been removed.

Page 16: Materials and methods

Section 3.1: Mention the reference of the methodology used for the preparation of yoghurt.

It has been added.

tank method in 85 ± 1 °C /20 min- It should be tank method for 85 ± 1 °C /20 min.

It has been  changed.

amount of nanocapsulated- Remove the space before nanocapsulated.

It was corrected.

After the oil addition, the milk was homogenized 3 times in Arm-field Ltd. Ringwood (England) homogenizer under a pressure of 7 MPa- What was the reason for homogenizing three times.

Thank you for this question. The 3-fold homogenisation ensures that the fat globules are thoroughly broken up and dispersed. The homogeniser used in this study is a low-capacity laboratory-scale apparatus mainly adapted to carry out effective homogenisation of milk fat globules in milk. Unfortunately, the homogenisation efficiency decreases when an additive such as vegetable oil rather than milk fat is used. Therefore, it was decided to repeat the process to ensure that no oil was released.

jogurt- Use the uniform terminology throughout the manuscript.

It was corrected.

based plate- DO you mean base plate ?

It was corrected.

tested samples was 4 ±1 °C.- It should be test samples.

It was corrected.

The films obtained were analysed in terms of nanoparticle size- Which films are being referred to?

It was corrected.

fabricated composite films- What does this mean ?

It was corrected.

Comments on the Quality of English Language: In the whole article, sentences are very large. It is suggested to make these sentences in small sentences for better understanding of the readers. The authors need to use proper words. 

Long sentences have been shortened or divided into several smaller ones.

Reviewer 2 Report

The authors compiled “Effect of nanoencapsulated walnut oil on the properties of yogurts and discussed about encapsulated of walnut oil and effect on yogurt quality. The manuscript is well written and provides fresh insight to researchers and industry professionals for functionalization of yogurt in different ways. Please address following points:

1. Grammatical/proof editing is required throughout the manuscript.

2. Words are combined at so many places, do the correction

3. Need to improve the abstract and, if possible, add some results.

4. Introduction section should be improved focus on after forticiation of yogurt properties with the latest references to clarify your study's intended meaning. 

5. Material and methods: it is possible if the author add images of yogurts before and after fortification.

6. Conclusion section.  The author needs to add some future recommendations at the end.  It is possible improvement sentences, It included repeated sentences.

It could be slighlty improvement the writing quality of paper.

Author Response

Authors: We thank the reviewer for valuable comments on the manuscript. All changes have been applied to the text using the change tracking function. We hope that the revision made has improved the manuscript and that its quality has improved significantly.

Reviewer:The authors compiled “Effect of nanoencapsulated walnut oil on the properties of yogurts” and discussed about encapsulated of walnut oil and effect on yogurt quality. The manuscript is well written and provides fresh insight to researchers and industry professionals for functionalization of yogurt in different ways. Please address following points:

  1. Grammatical/proof editing is required throughout the manuscript.

The manuscript has been revised according to the Reviewer's comments.

  1. Words are combined at so many places, do the correction

It was corrected.

  1. Need to improve the abstract and, if possible, add some results.

It was corrected.

  1. Introduction section should be improved focus on after forticiation of yogurt properties with the latest references to clarify your study's intended meaning. 

Introduction section was improved.

  1. Material and methods:it is possible if the author add images of yogurts before and after fortification.

The yoghurts were obtained in an identical manner, except that the nanoemulsion with walnut oil (described in chapter 3.2) was added to one batch of base milk and the milk powder alone to the other. Fermentation of both batches was then carried out. The resulting yoghurts did not differ visually, which was also confirmed in sensory tests. Photo in the appendix.

  1. Conclusion section.  The author needs to add some future recommendations at the end.  It is possible improvement sentences, It included repeated sentences.

It was added.

Comments on the Quality of English Language: It could be slightly improvement the writing quality of paper.

It was corrected.

Reviewer 3 Report

The manuscript entitled "Effect of nanoencapsulated walnut oil on the properties of yogurts" is a novel topic that could be very interesting for the readers. However, it should make major corrections as bellow.

1- Please modify the title of the article somehow to mention the type of the emulsion because it is one of the most important factors of the research.

2- please modify the abstract and mention some data of the work such as main analyses done as materials and methods. the structure of the abstract should be organized better.

3- the emulsion was prepared with skim milk powder ? the part Nanoemulsion of walnut oil preparation is misleading. please explain the emulsion preparation in more details was the mixture just skim milk powder and walnut oil?

4- the fortification of the yoghurt was done by (2 g/100 g) walnut oil. Where did you get this ratio? why did'nt you choose a range of concentrations to ensure the best ration in terms of physicochemical and rheological properties?

5- section 3.8. and 3.9. are talking about films??!! protocol has been copied and pasted from the references. please paraphrase it based on your own samples nature.

6- i section 2.6., the authors need to explain the effect of these particle sizes on the structure and physicochemical and rheological properties of the fortified yoghurts.

Author Response

Authors: We thank the reviewer for valuable comments on the manuscript. All changes have been applied to the text using the change tracking function. We hope that the revision made has improved the manuscript and that its quality has improved significantly.

Reviewer: The manuscript entitled "Effect of nanoencapsulated walnut oil on the properties of yogurts" is a novel topic that could be very interesting for the readers. However, it should make major corrections as bellow.

1- Please modify the title of the article somehow to mention the type of the emulsion because it is one of the most important factors of the research.

The title was modified according to Reviewer suggestion:

“An innovative method for the production of yoghurt fortified with walnut oil nanocapsules and characteristics of functional properties in relation to conventional yoghurts”.

2- please modify the abstract and mention some data of the work such as main analyses done as materials and methods. the structure of the abstract should be organized better.

The abstract has been reorganized.

3- the emulsion was prepared with skim milk powder ? the part Nanoemulsion of walnut oil preparation is misleading. please explain the emulsion preparation in more details was the mixture just skim milk powder and walnut oil?

It was corrected.

4- the fortification of the yoghurt was done by (2 g/100 g) walnut oil. Where did you get this ratio? why did'nt you choose a range of concentrations to ensure the best ration in terms of physicochemical and rheological properties?

Thank you very much for this question.This amount of oil addition was selected on the basis of previous studies conducted on kefirs (Turek & Wszołek 2021, Turek & Wszołek, 2022). Prior to the aforementioned studies, preliminary tests were performed, where the appropriate concentration of vegetable oil additive was selected based on the results of the sensory evaluation and fatty acid profile analysis. The current study with the addition of nano-encapsulated vegetable oil to dairy products is only in its preliminary phase. Certainly, additional studies will be carried out to check the most favourable oil addition on the mentioned physicochemical and rheological characteristics, not only in yoghurts but also in other dairy products.

5- section 3.8. and 3.9. are talking about films??!! protocol has been copied and pasted from the references. please paraphrase it based on your own samples nature.

It was corrected.

6- i section 2.6., the authors need to explain the effect of these particle sizes on the structure and physicochemical and rheological properties of the fortified yoghurts.

In this work, we focused on the generation of nanocapsules containing walnut oil by an innovative method, i.e. using milk powder as the encapsulating substance. The resulting fortified yoghurts were characterised, and the structural, physicochemical and rheological properties of the obtained fortified yoghurt were compared with conventional yoghurts. The results indicated that the presence of nano/microstructures had no significant effect on the mentioned properties. In our further studies, we focused on changing the concentration of added oil and investigating the effect of concentration on capsule size. Obtaining capsules of different sizes will enable us to analyse the effect of particle size on the structure and physicochemical and rheological properties of the enriched yoghurts.

Round 2

Reviewer 3 Report

Well done for addressing the comments.

Author Response

We thank the Reviewer for valuable comments on the manuscript. All changes have been applied to the text using the change tracking function. We hope that the revision made has improved the manuscript and that its quality has improved significantly.